# Plasma Metabolites and Liver Composition of Broilers in Response to Dietary *Ulva lactuca* with Ulvan Lyase or a Commercial Enzyme Mixture

**DOI:** 10.3390/molecules27217425

**Published:** 2022-11-01

**Authors:** Cristina M. Alfaia, Mónica M. Costa, Rui M. A. Pinto, José M. Pestana, Miguel Mourato, Patrícia Carvalho, Cátia F. Martins, Paula A. Lopes, Madalena M. Lordelo, José A. M. Prates

**Affiliations:** 1CIISA-Centre for Interdisciplinary Research in Animal Health, Faculdade de Medicina Veterinária, Universidade de Lisboa, 1300-477 Lisboa, Portugal; 2Laboratório Associado para Ciência Animal e Veterinária (AL4AnimalS), Faculdade de Medicina Veterinária, Universidade de Lisboa, 1300-477 Lisboa, Portugal; 3iMed.UL, Faculdade de Farmácia, Universidade de Lisboa, Avenida Professor Gama Pinto, 1649-003 Lisboa, Portugal; 4JCS-Laboratório de Análises Clínicas Dr. Joaquim Chaves, Rua Aníbal Bettencourt, n° 3, Outurela, 2790-224 Carnaxide, Portugal; 5LEAF-Linking Landscape, Environment, Agriculture and Food, Instituto Superior de Agronomia, Universidade de Lisboa, 1349-017 Lisboa, Portugal

**Keywords:** *Ulva lactuca*, carbohydrase, plasma profile, hepatic composition, poultry

## Abstract

The effect of a high incorporation level of *Ulva lactuca*, individually and supplemented with a Carbohydrate-Active enZyme (CAZyme) on broilers’ plasma parameters and liver composition is assessed here. Twenty one-day-old Ross 308 male broilers were randomly assigned to one of four treatments (*n* = 10): corn/soybean meal based-diet (Control); based-diet with 15% *U. lactuca* (UL); UL diet with 0.005% of commercial carbohydrase mixture; and UL diet with 0.01% of recombinant ulvan lyase. Supplementing *U. lactuca* with the recombinant CAZyme slightly compromised broilers’ growth by negatively affecting final body weight and average daily gain. The combination of *U. lactuca* with ulvan lyase also increased systemic lipemia through an increase in total lipids, triacylglycerols and VLDL-cholesterol (*p* < 0.001). Moreover, *U. lactuca*, regardless of the CAZyme supplementation, enhanced hepatic *n*-3 PUFA (mostly 20:5*n*-3) with positive decrease in *n*-6/*n*-3 ratio. However, broilers fed with *U. lactuca* with ulvan lyase reduced hepatic α- and γ-tocopherol concentrations relative to the control. Conversely, the high amount of pigments in macroalga diets led to an increase in hepatic β-carotene, chlorophylls and total carotenoids. Furthermore, *U. lactuca*, alone or combined with CAZymes, enhanced hepatic total microminerals, including iron and manganese. Overall, plasma metabolites and liver composition changed favorably in broilers that were fed 15% of *U. lactuca*, regardless of enzyme supplementation.

## 1. Introduction

Poultry meat is expected to represent 41% of all the protein meat sources by 2030 [1]. Macroalgae, due to their high growth rates and richness in nutritive and bioactive compounds, have emerged as sustainable alternative resources to conventional feedstuffs (i.e., corn and soybean) that could hinder the existing food-feed competition for grains or legumes [2]. Green macroalgae species, mostly *Ulva* sp., are rich in carbohydrates (up to 65% of dry matter, DM) [3], antioxidant chlorophylls and carotenoids [4] They are also rich in minerals including calcium, magnesium and iron, and have a balanced sodium and potassium proportion [5].

The protein content of *Ulva* sp. is variable (4.8–41.8% DM) [2], but with a good nutritional quality [5]. However, lipids are present in low amounts in alga biomass (<6.6% DM), although they contain health-promoting polyunsaturated fatty acids (PUFA) [6]. Moreover, ulvan is a sulphated soluble heteropolysaccharide, which is the major compound of the *Ulva* sp. cell wall (up to 30% DM) [7], followed by insoluble cellulose. This polymer has various bioactive functions as it can act as an anti-tumor, anti-coagulant, antihyperlipidemic, hepato-protective and immuno-stimulating agent [7].

The inclusion of *Ulva* sp. in poultry diets can influence animal growth performance and plasma biochemical parameters [8,9,10,11], with some studies showing a positive effect on the latter, such as the decrease of serum cholesterol [8]. However, the level of macroalga used in these reports was only up to 3–3.5% [8,10,11] and 6% [9] in the feed. In fact, although there are sustainable seaweed production methods (i.e., Integrated Multitrophic Aquaculture, ITMA), macroalga cultivation technology is still in development in order to reduce its production costs and environmental impact [2]. In addition, some studies suggested that the maximum level of *Ulva* sp. incorporated in a chicken diet should not surpass 10% in the feed, due to the presence of cell wall polysaccharides that are indigestible for monogastric diets with consequent impairment of nutrient digestibility [12,13].

Feed enzymes appear as a solution, since Carbohydrate-Active enZymes (CAZymes) were previously supplemented in broiler chicken diets for their ability to degrade intricate matrices of carbohydrates present in microalgae [14,15], macroalgae such as *Laminaria digitata* [16,17] and *Ulva* sp. [18,19,20] or grains [21,22]. Recently, an in vitro study showed the ability of a recombinant ulvan lyase from family 25 of polysaccharide lyases (PL25) to partially disrupt the *U. lactuca* cell wall and, thus, release mono- and oligosaccharides and some monounsaturated fatty acids (MUFA) (e.g., 18:1*c*9) [23].

The objective of the present study was to test if *U. lactuca,* when replacing 15% of corn and soybean meal, combined or not with a commercial carbohydrase mixture or recombinant ulvan lyase (PL25), would enhance a broiler’s metabolic state and relate that with results of growth performance previously discussed in a companion report [18]. Thus, the effects of *U. lactuca*, supplemented or not with CAZymes, on a broiler’s diet on plasma biochemical parameters and liver composition, including lipids, antioxidant pigments and mineral profile, were assessed.

## 2. Results

### 2.1. Feed Intake and Animal Growth Performance

Table 1 shows the effect of the experimental diets on the zootechnical parameters of broilers. Supplementing *U. lactuca* with the recombinant ulvan lyase decreased the final body weight (*p* = 0.016) relative to the control. Similarly, body weight gain decreased by 15.6 g/d in broilers that were fed macroalga supplemented with ulvan lyase (*p* = 0.018) in relation to the control. However, dietary treatments had no significant influence on the feed intake and the feed conversion ratio (*p* > 0.05).

### 2.2. Plasma Biochemical Profile

Plasma metabolites, key electrolytes such as chloride, sodium and potassium, and hepatic markers of broilers that were fed *U. lactuca*, supplemented or not with CAZymes, are presented in Table 2. Total lipids (*p* < 0.001), triacylglycerols (TAG) (*p* < 0.001) and VLDL-cholesterol (*p* < 0.001) were increased in broilers fed with *U. lactuca* supplemented with ulvan lyase compared to the other diets. Conversely, total cholesterol (*p* < 0.001) and LDL-cholesterol (*p* < 0.001) were reduced in broilers that were fed diets with *U. lactuca* alone or supplemented with commercial Rovabio^®^. Among all dietary treatments, HDL-cholesterol (*p* < 0.001) reached the highest values in broilers that were fed the control diet. Glucose (*p* = 0.013) and total protein (*p* < 0.001) were increased in broilers that were fed *U. lactuca* with the recombinant ulvan lyase diet relative to the other treatments. Urea (*p* = 0.102) was unaffected by dietary treatments, and residual creatinine concentrations (<0.001 mg/dL) were found for all treatments. The lowest value of the C-reactive protein (*p* < 0.001) was observed in broilers fed with *U. lactuca* alone. Regarding the ionogram, chloride (*p* < 0.001) and potassium (*p* < 0.001) concentrations were consistently higher in broilers that were fed *U. lactuca* with the recombinant CAZyme, when compared to the other groups. However, no differences were found between broilers fed with only *U. lactuca* or supplemented with commercial CAZyme (*p* > 0.05). In contrast, *U. lactuca*, with or without enzymes, increased sodium levels (*p* < 0.001) relative to the control. For the hepatic markers, macroalga containing diets, with or without CAZyme supplementation, reduced alanine aminotransferase (ALT) and aspartate aminotransferase (AST) levels (*p* < 0.001). In addition, broilers fed with *U. lactuca* alone or supplemented with commercial Rovabio^®^ had reduced alkaline phosphatase (ALP) levels (*p* < 0.001) in relation to the control, whereas gamma-glutamyltransferase (GGT) was highest (*p* < 0.001) in broilers that were fed *U. lactuca* supplemented with the recombinant CAZyme.

### 2.3. Hepatic Total Lipids, Cholesterol Contents and Fatty Acid Composition

Data on hepatic total lipids, total cholesterol and fatty acid composition of broilers that were fed *U. lactuca*, with or without CAZyme supplementation, are shown in Table 3. Total lipids were unaffected by dietary treatments (*p* = 0.086). In contrast, diets with macroalga increased total cholesterol relative to the control (*p* = 0.004). Overall, the most predominant fatty acids were 18:0 (25.9–27.2%), 18:2*n*-6 (20.6–24.1%), 16:0 (13.9–16.6%), 20:4*n*-6 (15.4–16.2%) and 18:1*c*9 (5.78–7.59%). Differences on the fatty acid profile were observed on 20 out of 27 fatty acids identified. *U. lactuca*, regardless of CAZyme supplementation, decreased the percentage of 16:0 (*p* < 0.001) relative to the control. On the other hand, minor saturated fatty acids (SFA), such as 15:0 (*p* < 0.001) and 17:0 (*p* < 0.001), were increased in the liver of broilers fed with *U. lactuca*, supplemented or not with feed enzymes and 20:0 (*p* < 0.001) and 22:0 (*p* < 0.001) in those fed the macroalga alone. Regarding monounsaturated fatty acids, the percentage of 16:1*c*7 (*p* < 0.001) was increased in broilers fed with *U. lactuca* alone or supplemented with the commercial Rovabio^®^, whereas the percentage of 16:1*c*9 (*p* = 0.001) and 18:1*c*9 (*p* = 0.001) decreased in broilers fed with *U. lactuca* supplemented with ulvan lyase. Residual values of 17:1*c*9 (*p* = 0.007) and 18:1*c*11 (*p* < 0.001) were observed in broilers that were fed the control diet. The percentages of the polyunsaturated fatty acids (PUFA) 18:2*n*-6 (*p* < 0.001), 18:3*n*-3 (*p* < 0.001), 20:3*n*-3 (*p* = 0.001), 20:5*n*-3 (eicosapentaenoic acid, EPA) (*p* < 0.001), 22:5*n*-3 (docosapentaenoic acid, DPA) (*p* < 0.001) and 22:6n-3 (docosahexaenoic acid, DHA) (*p* < 0.001) were consistently increased by the addition of *U. lactuca*, with or without the CAZyme supplementation. In particular, EPA and DHA were, respectively, 4.5- and 3-fold higher in the liver of broilers that were fed *U. lactuca*, alone or combined with feed enzymes, than in those fed the control diet. Total hepatic SFA (*p* < 0.001) and *n*-6/*n*-3 PUFA ratio (*p* < 0.001) were positively decreased by *U. lactuca*, with or without CAZymes. An inverse trend was observed for the sum of PUFA (*p* < 0.001), *n*-6 PUFA (*p* < 0.001), *n*-3 PUFA (*p* < 0.001) and the PUFA/SFA ratio (*p* < 0.001). However, the percentage of *cis*-MUFA (*p* = 0.003) was lowest in the liver of broilers fed with *U. lactuca* supplemented with the recombinant ulvan lyase.

### 2.4. Hepatic Vitamin E, Pigments and Mineral Profile

Hepatic vitamin E homologues, pigments and mineral composition of broilers that were fed *U. lactuca*, with or without CAZyme supplementation, are presented in Table 4. α-Tocopherol (*p* = 0.013) and γ-tocopherol (*p* = 0.009) contents were reduced in broilers fed with *U. lactuca* supplemented with the recombinant ulvan lyase compared to the control diet. In contrast, liver β-carotene (*p* < 0.001), chlorophyll-*a* (*p* < 0.001), chlorophyll-*b* (*p* < 0.001) and total carotenoids (*p* < 0.001) increased with *U. lactuca*, supplemented or not with CAZymes, relative to the control. Although macrominerals in the liver did not change across diets (*p* > 0.05), the most predominant were potassium and phosphorous with average values of 425 and 351 mg/100 g, respectively. In contrast, some trace elements, including iron and manganese, were affected by dietary treatments (*p* < 0.05). Iron (*p* = 0.001) was significantly higher in broilers that were only fed *U. lactuca*, intermediate in broilers that were fed *U. lactuca* combined with Rovabio^®^, and lower in those that were fed *U. lactuca* supplemented with ulvan lyase or the control diet, influencing in the same manner the total of microminerals (*p* = 0.001). Manganese levels (*p* = 0.001) were higher in broilers that were fed macroalga diets in comparison to the control.

### 2.5. Principal Component Analysis (PCA)

The variability of plasma parameters into two dimensions is shown in Figure 1. The first two factors (Figure 1A) account for 66% of the total variation (36.4% for factor 1 and 29.2% for factor 2). The control group was gathered in quadrant (a) while macroalga supplemented with the recombinant ulvan lyase was more dispersed in quadrants (b) and (d), but plainly discriminated from the other two groups. Data from *U. lactuca* alone or supplemented with the commercial Rovabio^®^ (UL and ULR) were located in quadrants (d) and (c) with no possible discrimination between them. These groups appear to overlap suggesting that the plasma biochemical profiles of these dietary treatments were partially similar. The loadings for the first two factors are presented in Figure 1B and Table 5. The plasma parameters with the highest discriminant power were chloride (0.905), TAG and VLDL-cholesterol (0.884), sodium (0.851) and GGT (0.818) for factor 1, and AST (0.828), ALT (0.826), total cholesterol (0.796), ALP (0.772), HDL-cholesterol (0.754) and LDL-cholesterol (0.724) for factor 2. The PCA performed with the hepatic chemical composition reveals a clear separation of the control diet but failed to discriminate macroalga treatments (see Appendix A).

## 3. Discussion

The majority of the studies carried out so far reported the dietary effect of low levels of *U. lactuca* (up to 3.5% feed) on broilers’ growth and plasma biochemical parameters [8,19]. Previously, we reported that the dietary incorporation of high levels (15%) of *U. lactuca*, especially when supplemented with ulvan lyase, slightly impaired broilers’ performance but improved meat composition through an accumulation of antioxidant carotenoids, *n*-3 PUFA and total minerals [18]. The negative effect of macroalga in animal performance was probably due to the presence of algal indigestible polysaccharides and high mineral content, which might have compromised the digestibility of feed compounds [24]. Although not significant, the tendency for a decrease in feed intake in broilers fed with *U. lactuca*-containing diets probably contributed to a reduction of average daily gain and final body weight, especially in those animals fed the diet supplemented with ulvan lyase. A numerical decrease in feed intake was also reported by Ventura et al. [12] when feeding increasing levels of *Ulva* sp. and, thus, it is possible that some algal compounds could have decreased feed palatability. Herein, we hypothesized that the addition of 15% *U. lactuca*, alone or in combination with both exogenous enzymes, might improve the plasma biochemical profile and change lipid metabolism in broilers. Most of the plasma metabolites were largely affected by the experimental diets; however, the values are within the reference values reported for broilers [25,26]. Total lipids, TAG and VLDL-cholesterol were consistently increased in broilers that were fed *U. lactuca* with the recombinant ulvan lyase, followed by reduced HDL-cholesterol levels, pointing towards a negative impact on the lipemia profile. In turn, *U. lactuca* was responsible for a reduction in total and LDL-cholesterol, which agrees with the findings of some works using 1 to 3% [27] and 2 to 6% [9] of *U. lactuca*, respectively, in broilers’ diet. It was reported higher total cholesterolemia levels (183.8 mg/dL) [25] than the ones found in our study, probably, due to the use of chicken layers instead of broilers. Total cholesterol, TAG, HDL-cholesterol and LDL-cholesterol have been used as key indicators of lipid metabolism balance [28,29]. The reverse cholesterol transport is the mechanism by which the organism removes excess of cholesterol from peripheral tissues and delivers it to the liver, where it will be redistributed to other tissues or removed from the organism, being HDL-cholesterol the main lipoprotein responsible for this process, whereas LDL-cholesterol has the opposite function [30]. Moreover, the majority of fatty acids is synthesized in the liver and transported via LDL for storage as triacylglycerols in the adipose tissue.

Broilers fed *U. lactuca* with or without the recombinant ulvan lyase displayed an increased glycemic response. Even if the levels were higher than the ones reported elsewhere [25], this small increment is thought to be devoid of clinical physiological relevance. The same rational applies for total protein. Unaffected renal function was proven by the non-variations of urea and creatinine across the experimental groups. The acute phase C-reactive protein (CRP) is produced by the liver and secreted into the blood in response to inflammation [31]. Herein, we found that CRP was lower in broilers fed *U. lactuca*, intermediate if supplemented with Rovabio^®^ Excel AP and recombinant ulvan lyase, and higher in the control group, pointing towards a beneficial anti-inflammatory effect of this macroalga. Notwithstanding, the highest value of CRP obtained for animals fed the control diet (0.026 mg/dL) was still below the normal averaged value found in the plasma of 35-d-old broilers (14 mg/dL) [32]. Sodium, potassium and chloride are electrolytes positively affected by dietary *U. lactuca*. While chloride and potassium presented a similar variation, with the highest levels in broilers fed *U. lactuca* with recombinant ulvan lyase, sodium was increased by *U. lactuca*, with or without enzymes. Potassium and sodium help the body maintaining fluid and blood volume so it can function normally [33]. Sodium is responsible for the regulation of the membrane potential of cells and, along with potassium, is exchanged across cell membranes as part of active transport. Chloride is an anion occurring predominantly in the extracellular fluid whose serologic levels are mostly regulated by the kidney [33]. AST, ALT and ALP were reduced by *U. lactuca*, regardless of the CAZyme supplementation. These changes on aminotransferases activity are within the reference values reported for birds using both micro- and macroalga species, *Arthrospira platensis* [14] and *L. digitata* [17], respectively. GGT levels were higher in broilers fed *U. lactuca* alone and supplemented with the recombinant ulvan lyase than the control diet, in resemblance to the variations observed previously by some authors [14,17], yet not implying any kind of liver injury or dysfunction.

Liver is the anatomical site for cholesterol synthesis and fatty acid oxidation in broilers. De novo lipogenesis, a highly regulated metabolic pathway, occurs in the liver but also in the adipose tissue [34]. Hepatic total lipids did not change by dietary incorporation of 15% *U. lactuca*, supplemented or not with exogenous feed enzymes, conversely to the plasma lipid profile. However, the addition of macroalga at this level in broilers’ diet promoted an increase of total cholesterol concentration in the liver, contrasting with our own results obtained with the dietary incorporation of 15% *L. digitata* [17]. Also, the dietary inclusion of the microalga *Chlorella vulgaris* at 10% level, alone or supplemented with CAZymes, had no effect on total cholesterol content in the liver of finishing pigs [35]. This discrepancy observed in different trials in response to total cholesterol in liver could be partly attributed to the dose and origin of the alga. Moreover, an effect of algal carotenoids, particularly xanthophylls, on increasing blood cholesterol levels might have occurred. This phenomenon was previously described in rodents [36,37], and then suggested when feeding the animals with extracted lipids from the brown seaweed, *Undaria pinnatifida* [38]. However, this mechanism of action is still unclear, and its occurrence in broilers lacks evidence.

Major variations were observed in hepatic *n*-3 fatty acids of broilers fed *U. lactuca*, supplemented or not with CAZymes. These functional fatty acids are of particular interest in animal feeds due to their anti-microbial and antioxidant properties [39]. The enhancement of *n*-3 PUFA and their antioxidant properties in the liver have been associated with the downregulation of PUFA oxidation-related genes expression and mitigation of lipid peroxidation [40]. The percentage of α-linolenic acid (18:3*n*-3), EPA and DHA increased in the liver about 2-fold, 4.5-fold and 3-fold, respectively, in broilers fed *U. lactuca*, with or without CAZyme supplementation, relative to the control. Also, *U. lactuca* alone or in combination with the commercial Rovabio^®^ increased the amounts of 18:4*n*-3. The raise in 18:3*n*-3 and 18:4*n*-3 is in line with the composition of the diets, but EPA was only 1.3-fold higher in macroalgae-containing diets than in the control and DHA was not even present in the diets (Appendix A. Therefore, the increase of the sum *n*-3 PUFA, and particularly of *n*-3 long-chain PUFA, in the liver of broilers fed *U. lactuca*, alone or supplemented with exogenous CAZymes, could be explained by de novo lipogenesis through the intake of the precursor 18:3*n*-3 in the biosynthetic pathway of *n*-3 long-chain PUFA [41]. These findings corroborate those of Costa et al. [17], who also documented an increase in hepatic *n*-3 fatty acids, as well as a decrease of *n*-6/*n*-3 ratio, with the incorporation of 15% *L. digitata*, with or without feed enzymes, to broiler diets. Similarly, Costa et al. [18] reported an increase in meat *n*-3 PUFA, including EPA, DPA and DHA, with the addition of *U. lactuca* combined with ulvan lyase. In contrast, the combination of *U. lactuca* and the recombinant CAZyme decreased *cis*-MUFA, particularly 18:1*c*9, which is not in agreement with the in vitro release of MUFA from *U. lactuca* biomass [23].

The impact of 15% *U. lactuca* incorporation in broilers’ diet, with or without feed CAZymes, on hepatic levels of tocopherols and pigments was also explored. α-Tocopherol is the major vitamin E compound with the highest antioxidant activity [42]. The combination of *U. lactuca* and ulvan lyase reduced both α- and γ-tocopherol contents compared to the control. Recently, a decrease of hepatic α-tocopherol was reported with 15% of dietary *L. digitata* and the commercial carbohydrase Rovabio^®^ [17]. Similarly, a reduction of α-tocopherol was observed in the meat of broilers fed 15% of *U. lactuca* [18]. Explaining the present results remains a challenge, since literature documenting the effect of macroalgae incorporation in broiler diets on hepatic vitamin E is scarce [17]. There is a high variability of vitamin E levels between and within macroalgae species [2], which, therefore, compromises the amount of vitamin deposited in the liver. The chemical composition of macroalgae biomass varies not only between species, but also with location and season of cultivation and maturity of the algae [43]. However, the concentration of α-tocopherol found in *U. lactuca* biomass was 0.793 mg/kg DM, which is below the value previously reported for *Ulva intestinalis* (8.8 mg/kg DM) [44]. In contrast, a consistent increase of the sum of carotenoids, including β-carotene, and chlorophylls-*a* and -*b* was observed in the liver with macroalgae treatments relative to the control, which is supported by the 72 and 23-fold higher total chlorophylls and carotenoids, respectively, in *U. lactuca*-containing diets. In general, *Ulva* sp. are rich in carotenoids, such as β- and α-carotenes and xanthophylls, and chlorophylls [4]. Carotenoids are natural lipophilic compounds that have been used in animal feed due to their potential to modify meat and yolk color and health benefits including antioxidant, antimicrobial, anti-inflammatory and anticarcinogenic activities [45,46]. The antioxidant potential of chlorophylls is less studied, although there is evidence of their ability to scavenge peroxyl radicals with a synergistic effect with vitamin E [46].

Regarding trace elements, the incorporation of 15% *U. lactuca* in broiler diets was not enough to promote major modifications on hepatic mineral concentrations, despite their high amounts in this microalga. Only iron and manganese, as well as total microminerals, were influenced by dietary inclusion of macroalga. These trace metals play an important role in metabolic processes by acting as cofactors of antioxidant enzymes [47]. Unfortunately, iodine and bromine were not determined in the present study, in spite of a high bioaccumulation of these elements by *U. lactuca* reaching up to 45.1 mg/kg DM of iodine and 694 mg/kg DM of bromine. However, other studies reported an increase of these minerals in the meat of broilers fed 15% of *L. digitata* {16] or *U. lactuca* [18], although within safe levels for human health. A similar result occurred in piglets fed 10% *L. digitata* [48]. In further studies, these minerals should be analyzed due to the well-known benefits of iodine in the general metabolism [49] and the toxicity of bromine [50].

## 4. Materials and Methods

### 4.1. Experimental Design, Diets and Samples Collection

The procedures agreed with the Ethics Commission of CIISA/FMV and ORBEA/ISA (protocol code number 1/ORBEA-ISA/2020, date of approval 7 July 2020), were accepted by the Animal Care Committee of National Veterinary Authority (Direção Geral de Alimentação e Veterinária, Lisboa, Portugal) and followed ARRIVE Guidelines for in vivo trials. One hundred and twenty 1-d-old male Ross 308 broiler chicks with an average body weight of 39.8 ± 0.20 g, vaccinated against Marek disease and infectious bronchitis, were individually tagged, and 3 broilers were housed per pen in a total of 40 wired-floor cages, with 10 replicate pens per treatment, for 35 d. The animals were maintained in an environmentally controlled room under standard brooding practices, with constant light. Room temperature and ventilation (relative humidity) were monitored continuously, following the guidelines for the Ross 308 broiler strain. The number of animals used in the experiment was chosen to follow the 3R´s principle [14,15]. After an adaptation period of 21 d with a corn and soybean meal-based diet, broilers received one of the 4 experimental diets for 14 d until the standard slaughter age of 35 d, as follows: (1) a corn-soybean meal based diet (Control); (2) a diet with 15% of *U. lactuca* powder (Algolesko; Plobannalec-Lesconil, Brittany, France) (UL); (3) the UL diet supplemented with 0.005% of a commercial CAZyme mixture (Rovabio^®^ Excel AP; Adisseo, Antony, France), containing endo-1,3(4)-β-glucanase and endo-1,4-β-xylanase activities (ULR); and (4) the UL diet supplemented with 0.01% of ulvan lyase, which is a recombinant enzyme from the family 25 of polysaccharide lyases (ULL) that is able to cleave the glyosidic bond between 3-sulfated rhamnose and D-glucuronic acid or L-iduronic acid residues in ulvan [23]. All diets were formulated to be isocaloric and isonitrogenous. The details about dietary ingredients are provided in Table 6. The chemical composition of *U. lactuca* and diets is shown in Appendix A.

Feed was provided daily, and broilers were weighed weekly. Feed intake, weight gain, and the feed conversion ratio were determined weekly. At the end of the experiment, one broiler per pen was slaughtered using electrical stunning and exsanguination. Blood and liver samples were collected from each of the 10 animals per experimental group, vacuum packed and stored at −20 °C.

### 4.2. Recombinant Ulvan Lyase Production

The recombinant gene was expressed in *Escherichia coli* (BL21) cells using the NZY auto-induction LB medium (Nzytech, Lisbon, Portugal) after overnight incubation at 140 rpm and 19 °C [23]. Ultrasonication was applied to disrupt cells followed by centrifugation. The protein extract was freeze-dried and incorporated at 0.01% in the ULL diet.

### 4.3. Chemical Analysis of U. lactuca and Diets

The proximate composition and gross energy of the green macroalga and diets were analysed using standard methods [51], whereas fatty acid methyl esters (FAME), diterpenes (tocopherols and tocotrienols) and pigments (β-carotene, chlorophyll-a, chlorophyll-b and total carotenoids) and minerals were determined as previously reported [16,18].

### 4.4. Assessment of Hepatic Diterpenes, Pigments and Mineral Profile

Diterpenes pigments and total cholesterol in the liver were determined by High-Performance Liquid Chromatography [HPLC] after the direct saponification of samples. For HPLC procedure, a normal-phase silica column (Zorbax RX-Sil, 250 mm × 4.6 mm i.d., 5 μm particle size, Agilent Technologies Inc., Palo Alto, CA, USA) was used allowing fluorescence detection of tocopherols and tocotrienols, and UV-visible photodiode array detection of cholesterol and β-carotene [52]. The analysis of pigments was carried out following the procedure of Teimouri et al. [53] and quantified as suggested by Hynstova et al. [54].

The mineral profile of the liver was evaluated as previously reported [16,18], and expressed as mg/kg dry matter. Succinctly, all minerals, except for iodine and bromine, were determined using Inductively Coupled Plasma–Optical Emission Spectrometry (ICP-OES, iCAP 7200 duo Thermo Scientific, Waltham, MA, USA), after digestion of samples with concentrated nitric and hydrochloric acids reacting with hydrogen peroxide followed by dilution and filtration through 90 mm diameter-filter papers. The digestion conditions were as follows: 1 h until 95 °C and 1 h at 95 °C. Calibration curves were created with multi-element standards (SCP Science, Quebec, Canada, PlasmaQual S22) for quantification of the elements. The iodine and bromine were determined in the macroalga and diets using an inductively coupled plasma mass spectrometer (ICP-MS) (Thermo X series II, Thermo Fisher Scientific, Waltham, MA, USA) preceded by alkaline extraction with Tetramethylammonium hydroxide (TMAH) solution at 25% *v*/*v*. Afterwards, samples were spiked with chemical standards in a Heating Graphite Block System for 3 h at 90 °C, and centrifuged and filtered with 0.45 µm filters.

### 4.5. Assessment of Hepatic Total Lipid Content and Fatty Acid Profile

Total lipids, in duplicate, were extracted from lyophilised (−60 °C and 2.0 hPa, lyophilizator Edwards Modulyo, Crawley, UK) liver samples [55]. Afterwards, fatty acids were converted into FAME by combined alkaline and acidic transesterification reaction (Sukhija and Palmquist, 1988). The analysis of FAME was carried out through gas-chromatography (GC-FID HP7890A Hewlett-Packard, Avondale, PA, USA), using nonadecanoic acid (19:0) methyl ester as internal standard [15]. Fatty acids were identified by comparison with the reference standard (FAME mix 37 compounds, Supelco Inc., Bellefonte, PA, USA) and expressed as percentage of total fatty acids.

### 4.6. Analysis of Plasma Biochemical Parameters

Total cholesterol, HDL-cholesterol, LDL-cholesterol, TAG, phospholipids, total protein, urea, creatinine and glucose concentrations, AST, ALT, ALP and GGT, as well as the electrolytes chloride, sodium and potassium were analysed using an automated Modular Hitachi Analytical System (Roche Diagnostics, Mannheim, Germany), through diagnostic kits (Roche Diagnostics). The determination of VLDL-cholesterol and total lipids were calculated using the formulas reported in the literature [56,57].

### 4.7. Statistical Analysis

The Statistical Analysis System (SAS) program (SAS Institute Inc., Cary, NC, USA) was used to analyze data by ANOVA from Generalized Linear Model (GLM) adjusted to the Tukey—Kramer method (PDIFF option) for multiple comparisons of least squares means at *p* < 0.05. The statistical model considered either the cage (for feed intake and feed conversion ratio) or the broiler (for body weight, body weight gain, plasma metabolites and hepatic parameters), as the experimental units. Additionally, plasma parameters and hepatic lipid composition were subjected to principal component analysis (PCA) using the SPSS Statistics for Windows (IBM Corp. released 2020, version 27.0, Armonk, NY, USA) for data exploratory purposes and a comprehension of underlying trends in the data structure.

## 5. Conclusions

Overall, a high-level (15%) incorporation of *U. lactuca* in broiler diets induced beneficial changes in plasma lipids, particularly a hypocholesterolaemic effect. However, when combined with the recombinant CAZyme, it increased systemic lipemia with no effect on plasma cholesterol. The macroalga was also responsible for an enhancement of liver antioxidant capacity and metabolism through an increase of antioxidant pigments, *n*-3 fatty acids and total microminerals including iron and manganese. Although these results indicate possible the health benefits of *U. lactuca* for animals*,* caution must be taken towards the slight growth impairment in broilers fed macroalgae and the fact that recombinant ulvan lyase did not ameliorate the effects of such a high level of alga in the diet. Further investigations are required to evaluate the incorporation of *U. lactuca* at lower levels in poultry diets.

## Figures and Tables

**Figure 1 molecules-27-07425-f001:**
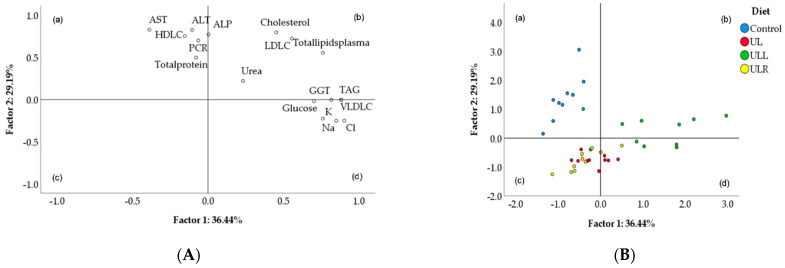
Loading plot of the first and second principal factors of the pooled data (**A**) and component score vectors (**B**) using plasma metabolites from broilers fed with different diets: Control, corn-soybean based-diet; UL, based-diet plus 15% *U. lactuca*; ULR, UL diet with 0.005% commercial CAZyme (Rovabio^®^ Excel AP); ULL, UL diet with 0.01% recombinant ulvan lyase; (a)–(d) represent the quadrants of two dimensional distribution of parameters.

**Table 1 molecules-27-07425-t001:** Effect of diets on growth performance of broilers (*n* = 10).

Parameters/Diets	Control	UL	ULR	ULL	SEM	*p*-Value
Initial weight (g)	767	748	754	727	82.5	0.763
Final weight (g)	1763 ^a^	1621 ^a,b^	1679 ^a,b^	1523 ^b^	159.7	0.016
Average daily gain (g/d)	77.4 ^a^	68.1 ^a,b^	72.1 ^a,b^	61.8 ^b^	3.38	0.018
Average daily feed intake (g/pen)	371	310	318	320	49.5	0.057
Feed conversion ratio	1.68	1.73	1.63	1.78	0.139	0.211

Diets: Control, corn-soybean based-diet; UL, based-diet plus 15% *U. lactuca*; ULR, UL diet with 0.005% commercial CAZyme (Rovabio^®^ Excel AP); ULL, UL diet with 0.01% recombinant ulvan lyase. SEM, standard error of the mean. ^a,b^ Different superscripts within a row differ significantly at *p* < 0.05.

**Table 2 molecules-27-07425-t002:** Plasma metabolites, electrolyte balance and hepatic markers of broilers as affected by diets (*n* = 10).

Parameters/Diets	Control	UL	ULR	ULL	SEM	*p*-Value
Plasma lipids						
Total lipids ^1^ (mg/dL)	429 ^b^	413 ^b,c^	400 ^c^	457 ^a^	7.35	<0.001
TAG (mg/dL)	37.5 ^b^	48.2 ^b^	43.6 ^b^	65.6 ^a^	3.78	<0.001
Total cholesterol (mg/dL)	121 ^a^	107 ^b^	103 ^b^	121 ^a^	2.45	<0.001
HDL-cholesterol (mg/dL)	82.4 ^a^	74.4 ^b^	75.2 ^b^	75.4 ^b^	1.24	0.001
LDL-cholesterol (mg/dL)	19.7 ^a^	15.2 ^b^	14.7 ^b^	21.7 ^a^	0.591	<0.001
VLDL-cholesterol ^2^ (mg/dL)	7.50 ^b^	9.64 ^b^	8.72 ^b^	13.1 ^a^	0.757	<0.001
Other plasma metabolites						
Glucose (mg/dL)	249 ^b^	254 ^a,b^	249 ^b^	258 ^a^	2.13	0.013
Urea (mg/dL)	2.00	1.57	2.09	1.92	0.152	0.102
Creatinine (mg/dL)	<0.001	<0.001	<0.001	<0.001	-	-
Total protein (g/dL)	2.37 ^b^	2.20 ^c^	2.34 ^b^	2.66 ^a^	0.028	<0.001
C-reactive protein (mg/dL)	0.03 ^a^	0.01 ^d^	0.01 ^c^	0.02 ^b^	0.001	<0.001
Ionogram						
Chloride (mEq/L)	109 ^c^	115 ^b^	116 ^b^	124 ^a^	1.07	<0.001
Sodium (mEq/L)	146 ^b^	151 ^a^	151 ^a^	155 ^a^	1.28	<0.001
Potassium (mEq/L)	7.13 ^c^	8.25 ^b^	8.15 ^b^	9.41 ^a^	0.171	<0.001
Hepatic markers						
ALT (U/L)	5.10 ^a^	3.20 ^b^	2.60 ^b^	3.50 ^b^	0.305	<0.001
AST (U/L)	531 ^a^	206 ^b^	183 ^b^	196 ^b^	23.9	<0.001
ALP (U/L)	1853 ^a^	1346 ^b^	1432 ^b^	1589 ^a,b^	70.6	<0.001
GGT (U/L)	18.5 ^c^	21.4 ^b^	18.2 ^c^	30.4 ^a^	0.589	<0.001

Diets: Control, corn-soybean based-diet; UL, based-diet plus 15% *U. lactuca*; ULR, UL diet with 0.005% commercial CAZyme (Rovabio^®^ Excel AP); ULL, UL diet with 0.01% recombinant ulvan lyase. SEM, standard error of the mean; TAG, triacylglycerols; HDL, high-density lipoproteins; LDL, low-density lipoproteins; VLDL, very low-density lipoproteins; ALT, alanine aminotransferase (EC 2.6.1.2); AST, aspartate aminotransferase (E.C. 2.6.1.1); ALP, alkaline phosphatase (EC 3.1.3.1); GGT, gamma-glutamyltransferase (EC 2.3.2.13). ^a,b,c,d^ Values within a row with different superscripts differ significantly at *p* < 0.05. ^1^ Total lipids = [total cholesterol] × 1.12 + [TAG] × 1.33 + 148; ^2^ VLDL-cholesterol = 1/5 [TAG].

**Table 3 molecules-27-07425-t003:** Hepatic total lipids and cholesterol contents and fatty acid composition of broilers as affected by diets (*n* = 10).

Parameters/Diets	Control	UL	ULR	ULL	SEM	*p*-Value
Total lipids (g/100 g)	2.69	2.44	2.47	2.43	0.079	0.086
Cholesterol (g/100 g)	3.07 ^b^	3.62 ^a^	3.55 ^a^	3.60 ^a^	0.114	0.004
Fatty acid composition (g/100 g FA)						
14:0	0.11	0.10	0.09	0.09	0.008	0.105
15:0	0.05 ^c^	0.09 ^a^	0.08 ^ab^	0.07 ^b^	0.003	<0.001
16:0	16.6 ^a^	14.5 ^b^	13.9 ^b^	14.4 ^b^	0.29	<0.001
16:1*c*7	0.29 ^b^	0.40 ^a^	0.38 ^a^	0.28 ^b^	0.022	0.001
16:1*c*9	0.14 ^a^	0.10 ^ab^	0.10 ^ab^	0.08 ^b^	0.010	0.009
17:0	0.33 ^b^	0.50 ^a^	0.45 ^a^	0.43 ^a^	0.018	<0.001
17:1*c*9	0.01 ^b^	0.02 ^a^	0.02 ^ab^	0.02 ^ab^	0.003	0.007
18:0	27.0	25.9	26.9	27.2	0.37	0.060
18:1*c*9	7.59 ^a^	7.09 ^ab^	6.36 ^bc^	5.78 ^c^	0.288	0.001
18:1*c*11	0.83 ^c^	1.42 ^a^	1.23 ^b^	1.12 ^b^	0.043	<0.001
18:2*n*-6	20.6 ^b^	24.1 ^a^	23.9 ^a^	23.8 ^a^	0.32	<0.001
18:3*n*-6	0.05 ^b^	0.08 ^a^	0.07 ^ab^	0.07 ^ab^	0.004	0.001
18:2*t*9*t*12	0.20	0.22	0.19	0.15	0.016	0.053
18:3*n*-3	0.07 ^b^	0.14 ^a^	0.14 ^a^	0.13 ^a^	0.006	<0.001
18:4*n*-3	0.04 ^b^	0.06 ^a^	0.06 ^a^	0.05 ^ab^	0.004	0.007
20:0	0.07 ^c^	0.21 ^a^	0.13 ^b^	0.13 ^b^	0.013	<0.001
20:1*c*11	0.20 ^ab^	0.25 ^a^	0.20 ^ab^	0.19 ^b^	0.014	0.012
20:2*n*-6	1.39	1.37	1.50	1.52	0.060	0.177
20:3*n*-6	1.98 ^a^	1.08 ^b^	1.32 ^b^	1.27 ^b^	0.121	<0.001
20:4*n*-6	15.4	15.7	16.1	16.2	0.39	0.434
20:3*n*-3	0.02 ^b^	0.04 ^a^	0.03 ^a^	0.03 ^a^	0.002	0.001
20:5*n*-3	0.03 ^b^	0.12 ^a^	0.13 ^a^	0.13 ^a^	0.012	<0.001
22:0	0.06 ^b^	0.14 ^a^	0.09 ^b^	0.08 ^b^	0.011	<0.001
22:1*n*-9	0.01	0.03	0.03	0.02	0.007	0.119
22:5*n*-3	0.25 ^b^	1.04 ^a^	1.16 ^a^	0.99 ^a^	0.063	<0.001
22:6*n*-3	0.86 ^b^	2.40 ^a^	2.52 ^a^	2.80 ^a^	0.167	<0.001
Others	5.88 ^a^	2.91 ^b^	2.99 ^b^	2.92 ^b^	0.247	<0.001
Partial sums of fatty acids (g/100 g FA)						
SFA ^1^	44.2 ^a^	41.4 ^b^	41.6 ^b^	42.4 ^b^	0.35	<0.001
*cis*-MUFA ^2^	9.07 ^a^	9.31 ^a^	8.31 ^ab^	7.49 ^b^	0.329	0.003
PUFA ^3^	40.8 ^b^	46.4 ^a^	47.1 ^a^	47.2 ^a^	0.34	<0.001
*n*-3 PUFA ^4^	1.27 ^b^	3.78 ^a^	4.04 ^a^	4.13 ^a^	0.206	<0.001
*n*-6 PUFA ^5^	39.4 ^b^	42.4 ^a^	42.9 ^a^	42.9 ^a^	0.29	<0.001
Ratios of fatty acids						
PUFA/SFA	0.92 ^b^	1.12 ^a^	1.13 ^a^	1.11 ^a^	0.014	<0.001
*n*-6*/n*-3	31.7 ^a^	11.6 ^b^	10.9 ^b^	10.6 ^b^	1.04	<0.001

Diets: Control, corn-soybean based-diet; UL, based-diet plus 15% *U. lactuca*; ULR, UL diet with 0.005% commercial CAZyme (Rovabio^®^ Excel AP); ULL, UL diet with 0.01% recombinant ulvan lyase. SEM, standard error of the mean; FA, fatty acids. ^a,b,c^ Values within a row with different superscripts differ significantly at *p* < 0.05. ^1^ SFA (saturated fatty acids) = Sum (12:0, 14:0, 15:0, 16:0, 17:0, 18:0, 20:0 and 22:0). ^2^ *cis*-MUFA (*cis*-monounsaturated fatty acids) = Sum (16:1*c*7, 16:1*c*9, 17:1*c*9, 18:1*c*9, 18:1*c*11, 20:1*c*11 and 22:1*n*-9). ^3^ PUFA (polyunsaturated fatty acids) = Sum (18:2*n*-6, 18:3*n*-6, 18:2*t*9*t*12, 18:3*n*-3, 18:4*n*-3, 20:2*n*-6, 20:3*n*-6, 20:4*n*-6, 20:3*n*-3, 20:5*n*-3, 22:5*n*-3 and 22:6*n*-3). ^4^ *n*-3 PUFA = Sum (18:3*n*-3, 18:4*n*-3, 20:3*n*-3, 20:5*n*-3, 22:5*n*-3 and 22:6*n*-3). ^5^ *n*-6 PUFA = Sum (18:2*n*-6, 18:3*n*-6, 20:2*n*-6, 20:3*n*-6 and 20:4*n*-6).

**Table 4 molecules-27-07425-t004:** Hepatic vitamin E, pigments and mineral contents of broilers as affected by diets (*n* = 10).

Parameters/Diets	Control	UL	ULR	ULL	SEM	*p*-Value
Diterpene profile (µg/g)						
α-Tocopherol	18.6 ^a^	16.3 ^ab^	15.1 ^ab^	12.6 ^b^	1.22	0.013
γ-Tocopherol	0.33 ^a^	0.27 ^ab^	0.30 ^ab^	0.22 ^b^	0.022	0.009
Pigments (µg/100 g)						
β-Carotene	0.26 ^b^	14.5 ^a^	14.7 ^a^	14.4 ^a^	1.67	<0.001
Chlorophyll-*a* ^1^	0.09 ^b^	0.46 ^a^	0.50 ^a^	0.43 ^a^	0.044	<0.001
Chlorophyll-*b* ^2^	0.14 ^b^	0.79 ^a^	0.78 ^a^	0.80 ^a^	0.069	<0.001
Total carotenoids ^3^	5.52 ^b^	143 ^a^	146 ^a^	146 ^a^	12.8	<0.001
Mineral profile (mg/100 g)						
Calcium	32.8	34.3	32.2	34.3	1.02	0.390
Magnesium	20.6	19.5	20.4	21.4	1.29	0.771
Phosphorous	353	347	350	353	3.90	0.699
Potassium	424	425	430	423	7.30	0.927
Sodium	99.2	94.6	92.7	98.3	2.93	0.385
Sulphur	231	242	240	240	4.00	0.284
Total macrominerals (M)	1161	1163	1188	1170	10.9	0.340
Copper	0.38	0.39	0.42	0.40	0.018	0.339
Iron	55.6 ^b^	88.3 ^a^	68.6 ^ab^	45.8 ^b^	6.76	0.001
Manganese	0.36 ^b^	0.47 ^a^	0.49 ^a^	0.46 ^a^	0.021	0.001
Zinc	2.54	2.67	2.49	2.59	0.094	0.565
Total microminerals (m)	53.3 ^b^	91.8 ^a^	72.0 ^ab^	49.3 ^b^	7.14	0.001
Total minerals	1214	1255	1257	1219	15.5	0.112

Diets: Control, corn-soybean based-diet; UL, based-diet plus 15% *U. lactuca*; ULR, UL diet with 0.005% commercial CAZyme (Rovabio^®^ Excel AP); ULL, UL diet with 0.01% recombinant ulvan lyase. SEM, standard error of the mean. ^a,b^ Values within a row with different superscripts differ significantly at *p* < 0.05. ^1^ Ca = 11.24 A662 – 2.04 A645. ^2^ Cb = 20.13 A645 – 4.19 A662. ^3^ Ca + b = 7.05 A662 + 18.09 A645.

**Table 5 molecules-27-07425-t005:** Loadings for the first two principal factors of plasma parameters.

Variables	Factor 1	Factor 2
Cholesterol	0.453	0.796
LDL-C	0.556	0.724
HDL-C	−0.154	0.754
VLDL-C	0.884	−0.002
TAG	0.884	−0.002
Total lipids	0.762	0.555
Glucose	0.702	−0.019
Urea	0.232	0.221
Total protein	−0.079	0.498
C-reactive protein	−0.064	0.700
Chloride	0.905	−0.250
Sodium	0.851	−0.251
Potassium	0.762	−0.226
ALT	−0.105	0.826
AST	−0.389	0.828
GGT	0.818	−0.002
ALP	0.004	0.772

TAG, triacylglycerols; HDL, high-density lipoproteins; LDL, low-density lipoproteins; VLDL, very low-density lipoproteins; ALT, alanine aminotransferase (EC 2.6.1.2); AST, aspartate aminotransferase (E.C. 2.6.1.1); ALP, alkaline phosphatase (EC 3.1.3.1); GGT, gamma-glutamyltransferase (EC 2.3.2.13).

**Table 6 molecules-27-07425-t006:** Main ingredients of the experimental diets.

Diet Composition	Control	UL	ULR	ULL
Ingredients (% as fed basis)				
Corn	50.4	43.7	43.7	43.7
Soybean meal	41.2	33.2	33.2	33.2
Sunflower oil	4.80	5.98	5.98	5.98
Sodium chloride	0.38	0.00	0.00	0.00
Calcium carbonate	1.10	0.00	0.00	0.00
Dicalcium phosphate	1.60	1.40	1.40	1.40
DL-Methionine	0.12	0.17	0.17	0.17
L-Lysine	0.00	0.12	0.12	0.12
Vitamin-mineral premix ^1^	0.40	0.40	0.40	0.40
*Ulva lactuca* powder	-	15.0	15.0	15.0
Rovabio^®^ Excel AP	-	-	0.005	-
Recombinant CAZyme	-	-	-	0.01

Diets: Control, corn-soybean based-diet; UL, based-diet plus 15% *U. lactuca*; ULR, UL diet with 0.005% commercial CAZyme (Rovabio^®^ Excel AP); ULL, UL diet with 0.01% recombinant ulvan lyase. ^1^ Premix provided the following nutrients per kg of diet: pantothenic acid 10 mg, vitamin D_3_ 2400 IU, cyanocobalamin 0.02 mg, folic acid 1 mg, vitamin K_3_ 2 mg, nicotinic acid 25 mg; vitamin B_6_ 2 mg, vitamin A 10,000 UI, vitamin B_1_ 2 mg, vitamin E 30 mg, vitamin B_2_ 4 mg, Cu 8 mg, Fe 50 mg, I 0.7 mg, Mn 60 mg, Se 0.18 mg, Zn 40 mg.

## Data Availability

Not applicable.

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
