# Peer review of "Plasma Metabolites and Liver Composition of Broilers in Response to Dietary Ulva lactuca with Ulvan Lyase or a Commercial Enzyme Mixture"

_molecules, 2022, doi:10.3390/molecules27217425_

Round 1
Reviewer 1 Report
The study presented in the manuscript evaluated dietary supplementation of 15% Ulva lactuca with or without enzyme on lipid metabolism in broilers. The Abstract and Introduction are sufficiently informative and complete. The objective of the study is clearly presented. The materials and methods described herein allow reproducibility, while the majority of results were interpreted appropriately and mostly discussed. Overall, the manuscript is well written and of high quality. It provides new information on the use of Ulva lactuca as a feed additive in poultry, although more investigation and further clarification are warranted.
Specific comments
One major finding of this study is that the weight gain of broilers was significantly decreased by the dietary use of 15% Ulva lactuca, especially by the ULL treatment. It is interesting that the incorporation of Ulva lactuca in the diet, regardless of the enzyme, tended to decrease the average daily feed intake (P = 0.057), which seems to be responsible for the compromised growth performance. Why did dietary Ulva lactuca negatively affect feed intake (the possible mechanism)? Does this have been observed in other studies? The negative influence of Ulva lactuca on growth performance should be discussed in the manuscript.
Author Response
- The study presented in the manuscript evaluated dietary supplementation of 15% Ulva lactuca with or without enzyme on lipid metabolism in broilers. The Abstract and Introduction are sufficiently informative and complete. The objective of the study is clearly presented. The materials and methods described herein allow reproducibility, while the majority of results were interpreted appropriately and mostly discussed. Overall, the manuscript is well written and of high quality. It provides new information on the use of Ulva lactuca as a feed additive in poultry, although more investigation and further clarification are warranted.
Reply: Thank you for your comment. We addressed your comments and suggestions below.
- Specific comments: One major finding of this study is that the weight gain of broilers was significantly decreased by the dietary use of 15% Ulva lactuca, especially by the ULL treatment. It is interesting that the incorporation of Ulva lactuca in the diet, regardless of the enzyme, tended to decrease the average daily feed intake (P = 0.057), which seems to be responsible for the compromised growth performance. Why did dietary Ulva lactuca negatively affect feed intake (the possible mechanism)? Does this have been observed in other studies? The negative influence of Ulva lactuca on growth performance should be discussed in the manuscript.
Reply: The negative effect of Ulva lactuca on broiler growth performance was probably caused by high levels of non-starch polysaccharides and minerals in the macroalga. The presence of algal indigestible polysaccharides compromises the digestibility of feed compounds, whereas the excess of minerals can increase the occurrence of diarrhoea in broilers. These aspects were previously described in recent studies (Bikker et al., 2020 https://doi.org/10.1016/j.anifeedsci.2020.114460; Stokvis et al., 2020 https://doi.org/10.1016/j.anifeedsci.2021.115061). In the present study, the tendency for a reduction in feed intake with dietary U. lactuca can be due to a decrease of feed palatability with the alga or even loss of appetite by broilers fed macroalgae-containing diets. Indeed, some animals presented diarrhoea but that was only severe for 3 of them (only 2.5% of mortality, as specified in Costa et al., 2022; https://doi.org/10.3390/ani12131720). Although other reports found a decrease of growth performance in chickens fed Ulva sp. (Matshogo et al., 2020 https://doi.org/10.3390/agriculture10110547; Alagan et al., 2020 https://doi.org/10.1186/s43088-020-00087-3; Ventura et al., 1994 https://doi.org/10.1016/0377-8401(94)90083-3), only Ventura et al., 1994 reported a numerical decrease of feed intake with increasing dietary levels of macroalgae (10 to 30% feed). This is probably explained by the low doses of Ulva sp. used in most of the studies (up to 3.5%) and the variety of nutritional composition found for these algae. The aspects related to broiler growth performance are now discussed in the manuscript between lines 214 and 222, page 7.
Reviewer 2 Report
Dear Authors,
The idea of the manuscript is interesting, the article is well written but I have some minor revision to be done:
Material and Methods:
Please describe the environmental conditions (light regimen, temperature, humidity, ventilation), the vaccination program.
Line 365: How many blood samples/group did you collect?
4.3. How did you express FAME?
Vitamin E: describe more the procedure, the expression of the results.
MINERAL profile : What minerals did you quantify? Describe more the procedure, the digestion method applied, the expression of the results.
Table 7: Use 2 or 3 decimals (not both) in all tables.
Line 413: To avoid the repetition of the analysis method, being the same for macroalgae, diet, liver, I think it is good if you present the method for all types of samples.
The conclusion is quite long, please review it presenting the most significant results and recommendations.
Author Response
Dear Authors,
- The idea of the manuscript is interesting, the article is well written but I have some minor revision to be done.
Reply: Thank you for the comments.
- Material and Methods:
- Please describe the environmental conditions (light regimen, temperature, humidity, ventilation), the vaccination program.
Reply: The environmental conditions are now specified between lines 340 and 342 (page 10).
- Line 365: How many blood samples/group did you collect?
Reply: The number of blood samples are now included between lines 363 and 364 (page 10).
- How did you express FAME?
Reply: The required information is now written between lines 380 and 381 (page 11).
- Vitamin E: describe more the procedure, the expression of the results.
Reply: The procedure is now described in more detail between lines 382 and 386 (page 11).
1.5. MINERAL profile: What minerals did you quantify? Describe more the procedure, the digestion method applied, and the expression of the results.
Reply: The quantified minerals were Br, Ca, Cu, I, Fe, Mg, Mn, P, K, Na and Zn for diets and macroalga (Table 7), and the same, except bromine and iodine, for liver (Table 4). The procedures applied were different between bromine and iodine and the rest of the minerals, and are now specified between lines 387 and 401 (page 11).
- Line 413: To avoid the repetition of the analysis method, being the same for macroalgae, diet, liver, I think it is good if you present the method for all types of samples.
Reply: The method for all types of samples is now described between lines 391 and 414 (page 11).
- Table 7: Use 2 or 3 decimals (not both) in all tables.
Reply: Two decimals are now used in all tables, except when there is two or more digits before the comma in which case only one decimal was used.
- The conclusion is quite long, please review it presenting the most significant results and recommendations.
Reply: The conclusion was reviewed, accordingly, between lines 464 and 484, pages 13 and 14.
Reviewer 3 Report
The paper (written in good English) presents partial results of one experiment with broiler chickens, but such information is not disclosed in the body of the manuscript. The Authors used maize-soybean meal-based (control) diet and maize-soybean meal diet containing as many as 15% of an expensive Ulva lactuca powder. The macroalga diet was prepared (and fed) in three versions: without feed enzymes, supplemented (on top) with ulvan lyase (a recombinant enzyme that cleaves the bond between 3-sulfated rhamnose linked to either D-glucuronic acid or L-iduronic acid in the marine polysaccharide ulvan) or supplemented (on top) with commercial enzyme preparation Rovabio® Excel AP of endo-1,3(4)-beta-glucanase and endo-1,4-beta-xylanase activities. The latter fact is not mentioned in the title. The effect of dietary treatments on liver and renal functions, electrolyte balance and inflammation parameters of blood plasma, and on contents of cholesterol, tocopherols, pigments, macrominerals, several trace minerals and lipid FA composition of liver tissue is presented in a total of 3 tables. Other experimental data, including results of chemical analyses of U. lactuca powder and diets (Table 7), and growth performance parameters of broilers (Table 1 in this manuscript), not mentioning considerable part of the description of M&M, are wholly copied from an article freshly published by the same authors' team in Animals MDPI (https://doi.org/10.3390/ani12131720). And, at least for me, this is a typical example of self-plagiarism.
On page 10 the Authors inform that “All diets were formulated to be isocaloric and isonitrogenous” (lines 352–353), but the macroalga diet(s) differed greatly from the control feed with regard of the amounts of n-3 FA (e.g. EPA ca 70-fold higher in U. lactuca diets), pigments (total carotenoids ca 23-fold higher in U. lactuca diets), as well as potassium and sodium. Therefore, it could have been easily predicted that macroalga diet(s), regardless of the presence of enzymes, would increase “hepatic n-3 PUFA (mostly 20:5n-3) with positive decrease of n-6/n-3 ratio” and that “hepatic beta-carotene, chlorophylls and total carotenoids [would be] were consistently higher in macroalga diets”(Abstract).
As regards Principal Component Analysis. If the number of samples amounted to 10 per feeding group (n = 10), what was the reason to perform the PCA (a technique for reducing the dimensionality of large datasets having thousands columns) with the banal conclusion that plasma biochemical indices (17 variables) of UL and ULR treatments were partially similar (lines 192–193) or that “clear separation of the control diet’’ existed for the (44) liver chemical components (lines 198–199), which quite evidently results from the data provided in Tables 2, 3 and 4? In my opinion, this is nothing more than making the manuscript more ‘scientific’ and making authors’ claims more convincing before the readers.
As a general rule, which probably is very well known to the Authors, manuscripts submitted to MDPI journals should only report results that have not been published before, even in part. In view of the above, the decision on the manuscript rests solely with the Editor(s).
Author Response
- The paper (written in good English) presents partial results of one experiment with broiler chickens, but such information is not disclosed in the body of the manuscript.
Reply: Thank you for your recommendation. That information is now disclosed in the body of the manuscript (lines 70 to 74, page 2).
- The Authors used maize-soybean meal-based (control) diet and maize-soybean meal diet containing as many as 15% of an expensive Ulva lactuca The macroalga diet was prepared (and fed) in three versions: without feed enzymes, supplemented (on top) with ulvan lyase (a recombinant enzyme that cleaves the bond between 3-sulfated rhamnose linked to either D-glucuronic acid or L-iduronic acid in the marine polysaccharide ulvan) or supplemented (on top) with commercial enzyme preparation Rovabio® Excel AP of endo-1,3(4)-beta-glucanase and endo-1,4-beta-xylanase activities. The latter fact is not mentioned in the title.
Reply: The use of high concentration of seaweed in the broiler diets is indeed expensive, despite a decrease of macroalgae cost is expectable. However, seaweeds can be sustainable alternatives to cereals as they do not compete with feed, food and fuel production and, although their cultivation technology is still in development in order to reduce production costs and ecological footprint, there are already sustainable seaweed production methods (i.e., Integrated Multitrophic Aquaculture, ITMA). These aspects are now written in detail between lines 38 and 40 (page 1) and lines 55 and 57 (page 2). The activities of the enzymes are specified between lines 361 and 365, page 10. The title was also modified, as suggested.
- The effect of dietary treatments on liver and renal functions, electrolyte balance and inflammation parameters of blood plasma, and on contents of cholesterol, tocopherols, pigments, macrominerals, several trace minerals and lipid FA composition of liver tissue is presented in a total of 3 tables. Other experimental data, including results of chemical analyses of lactuca powder and diets (Table 7), and growth performance parameters of broilers (Table 1 in this manuscript), not mentioning considerable part of the description of M&M, are wholly copied from an article freshly published by the same authors' team in Animals MDPI (https://doi.org/10.3390/ani12131720). And, at least for me, this is a typical example of self-plagiarism.
Reply: We understand the reviewer´s concern. However, we decided to keep Tables 1, 6 and 7 in the present manuscript because that data show what the composition of the experimental diets was and how diets influenced broilers´ growth performance. Therefore, it gives a context to the experiment and helps to understand better the experimental design and discuss the results obtained on liver and plasma parameters (please, see lines 216 to 219, page 7).
- On page 10 the Authors inform that “All diets were formulated to be isocaloric and isonitrogenous” (lines 352–353), but the macroalga diet(s) differed greatly from the control feed with regard of the amounts of n-3 FA (e.g. EPA ca 70-fold higher in U. lactuca diets), pigments (total carotenoids ca 23-fold higher in U. lactuca diets), as well as potassium and sodium. Therefore, it could have been easily predicted that macroalga diet(s), regardless of the presence of enzymes, would increase “hepatic n-3 PUFA (mostly 20:5n-3) with positive decrease of n-6/n-3 ratio” and that “hepatic beta-carotene, chlorophylls and total carotenoids [would be] were consistently higher in macroalga diets” (Abstract).
Reply: The amount of EPA was low for all diets and slightly differed between control and U. lactuca and diets (Table 7 was updated to present the correct values for fatty acid profiles and we apologize for our mistake). However, it is true that total pigments and minerals (mostly potassium and sodium) are considerably higher with U. lactuca diets than with control, but, when the diets were formulated, we just considered gross energy, crude fat and crude protein values to make them isocaloric and isonitrogenous. The relation between the high amount of pigments in macroalga diets and their increase in the liver is now clarified in the Abstract (lines 29 and 30). In addition, the discussion about liver fatty acids and pigments was modified between lines 300 and 303, 330 and 332 (pages 9 and 10).
- As regards Principal Component Analysis. If the number of samples amounted to 10 per feeding group (n = 10), what was the reason to perform the PCA (a technique for reducing the dimensionality of large datasets having thousands columns) with the banal conclusion that plasma biochemical indices (17 variables) of UL and ULR treatments were partially similar (lines 192–193) or that “clear separation of the control diet’’ existed for the (44) liver chemical components (lines 198–199), which quite evidently results from the data provided in Tables 2, 3 and 4? In my opinion, this is nothing more than making the manuscript more ‘scientific’ and making authors’ claims more convincing before the readers.
Reply: The reviewer as a point, since PCA is mostly used for parsimony (reduction of sample dimension) and higher sample sizes normally return more stable and consistent eigenvectors. However, in biological sciences, it is not always possible to have a high samples size because of several limitations concerning the acquisition and maintenance of animals. Therefore, we used PCA as a data exploratory rather than a hypothesis testing procedure, in order to comprehend data and to seek the underlying trends and gradients in the data structure (this aspect is now described in the manuscript between lines 487 and 488, page 14). Also, some studies observed that samples smaller than traditionally recommended are likely sufficient with high communalities. The recovery of population factors in sample data can be very good, almost regardless of sample size, level of overdetermination, or the presence of model error (Shaukat et al., 2016 http://doi.org/10.1515/eko-2016-0014; MacCallum et al., 1999 http://doi.org/10.1207/S15327906MBR3604_06). Please, see other publications using PCA with a low sample size (Lv & Zhao et al., 2007 https://doi.org/10.1007/s12161-016-0588-1; Campler et al. 2009 https://doi.org/10.1016/j.beproc.2008.12.018; Costa et al. 2022 https://doi.org/10.3390/ani12081007; Costa et al. 2022 https://doi.org/10.1186/s12917-022-03250-3; Coelho et al. 2022 https://doi.org/10.1016/j.rvsc.2022.01.008).
- As a general rule, which probably is very well known to the Authors, manuscripts submitted to MDPI journals should only report results that have not been published before, even in part. In view of the above, the decision on the manuscript rests solely with the Editor(s).
Reply: We acknowledged the reviewers’ concern. Please, see the response to comment 3.
Round 2
Reviewer 3 Report
Dear Authors,
The revised version of manuscript molecules-1943467 can be accepted for publication. However, I regret to say that all values pertaining FA profile presented now in Table 7, are quite different from those in the original manuscript. This creates suspicion (close to certainty) about manipulation of these data. What is more, I’m wondering why the FA profile data in the original Table 7, in the Costa et al. 2022 (Ulva lectuca, Animals, 12, 1720; Table 2) and in the Costa et al. 2022 (Laminaria digitata, BMC Veterinary Research, 18:153; Table 8) are the exact same (to the letter)? Well, the thing to remember before reading ‘scientific’ articles authored by your ‘pre-eminent’ team.
All the best.
Author Response
Thank you for recommending the acceptance of our paper and your observation. We apologize for our mistake. Indeed, we wrote the papers about the use of Ulva lactuca in broilers almost at the same time as the ones concerning Laminaria digitata and, therefore, we, erroneously, copied the fatty acid values from Laminaria´s to Ulva´s papers. We have already written an Erratum to correct our mistake in Animals´ Editor. We hope that our mistake will not tarnish our reputation and we appreciate your consideration. In attachment, you can find the request for modification of paper “Ulva lactuca, Animals, 12, 1720; Table 2” that we previously sent to the Editor.
